# Identifying subgroups with differential levels of service response to a digital screening and service navigation program for unmet social care needs

James R. John[1,2,3☯], Teresa Winata[1,4☯], Si Wang[5], Melissa Smead[6], Weng Tong Wu[1], Jane Kohlhoff[1,7], Virginia Schmied[8], Bin Jalaludin[9,10], Kenny Lawson[11], Siaw-Teng Liaw[10], Raghu Lingam[12], Andrew Page[11], Christa Lam-Cassettari[1,2,3], Katherine Boydell[1,13], Ping-I. Lin[1,14], Ilan Katz[15], Ann Dadich[11], Shanti Raman[9], Rebekah Grace[16], Aunty Kerrie Doyle[11], Tom McClean[17], Blaise Di Mento[1], John Preddy[6,18‡], Susan Woolfenden[19,20‡], Valsamma Eapen[1,2,3‡]*

1 School of Clinical Medicine, Faculty of Medicine and Health, University of New South Wales, Sydney, New South Wales, Australia, 2 Ingham Institute for Applied Medical Research, Liverpool, New South Wales, Australia, 3 Academic Unit of Infant, Child and Adolescent Psychiatry, South Western Sydney Local Health District, Liverpool, New South Wales, Australia, 4 National Disability Insurance Scheme Quality and Safeguards Commission, Parramatta, New South Wales, Australia, 5 Research and Evaluation Group, The Salvation Army, Sydney, New South Wales, Australia, 6 Murrumbidgee Local Health District, Wagga Wagga, New South Wales, Australia, 7 Karitane, Carramar, New South Wales, Australia, 8 School of Nursing and Midwifery, Western Sydney University, Parramatta, New South Wales, Australia, 9 South Western Sydney Local Health District, Liverpool, New South Wales, Australia, 10 School of Population Health, Faculty of Medicine and Health, University of New South Wales, Sydney, New South Wales, Australia, 11 School of Medicine, Western Sydney University, Sydney, New South Wales, Australia, 12 Population Child Health Research Group, School of Women's and Children's Health, Faculty of Medicine, University of New South Wales, Sydney, New South Wales, Australia, 13 Black Dog Institute, Sydney, New South Wales, Australia, 14 Department of Psychiatry and Behavioral Neuroscience, School of Medicine, Saint Louis University, Missouri, United States of America, 15 Social Policy Research Centre, Faculty of Arts, Design, and Architecture, University of New South Wales, Sydney, New South Wales, Australia, 16 Transforming early Education and Child Health Research Centre, Western Sydney University, Penrith, New South Wales, Australia, 17 Uniting NSW.ACT, Sydney, New South Wales, Australia, 18 Rural Clinical School, School of Clinical Medicine, University of New South Wales, Wagga Wagga, New South Wales, Australia, 19 Sydney Medical School, Faculty of Medicine and Health, University of Sydney, Sydney, New South Wales, Australia, 20 Sydney Local Health District, Sydney, New South Wales, Australia

☯ These authors contributed equally as joint first authors
‡ These authors contributed equally as joint senior authors
* v.eapen@unsw.edu.au

## Abstract

### Background

Digital screening and navigation interventions are increasingly integrated into health systems to identify and support families' unmet social care needs, yet their effectiveness in improving outcomes remains unclear among priority population communities. We hypothesise that responses to such digital interventions might vary based on sociodemographic and psychosocial characteristics.

which permits unrestricted use, distribution, and reproduction in any medium, provided the original author and source are credited.

**Data availability statement:** The individual level data collected and analysed during the current study are not publicly available due to ethical and privacy reasons. Data sharing has been restricted by the South Western Sydney Local Health District Human Research Ethics Committee. Requests will be reviewed in accordance with the approval of the South Western Sydney Local Health District Human Research Ethics Committee (swslhd-ethics@health.nsw.gov.au) to ensure that all ethical and legal requirements for data protection are met.

**Funding:** This study was supported through the NSW Health COVID-19 Research Grants Round 2. The funding body had no role in the design of the study, data collection, analysis, interpretation, or the writing of this manuscript.

**Competing interests:** The authors have declared that no competing interests exist.

## Methods

Data were analysed from 288 participants in a randomised controlled trial evaluating Watch Me Grow-Electronic – a digital screening and service navigation model to identify psychosocial needs, parental wellbeing, and child developmental needs in South Western Sydney (urban site) and Murrumbidgee (regional/rural site), New South Wales, Australia. Latent class analysis was used to identify subgroups of families based on parental and child clinical and sociodemographic factors. A zero-inflated negative binomial regression was conducted to assess changes in unmet needs, stratified by class and intervention group.

## Results

Three distinct classes were identified. Class 1 ($n = 134$) included people who were entirely non-culturally and linguistically diverse (CALD) background, in good mental health, with higher education and socioeconomic status (SES), and from the regional/rural site. Class 2 ($n = 94$) included people who were predominantly non-CALD, of low education and SES, had poor mental health, and from the regional/rural site. Class 3 ($n = 56$) included people of CALD, high SES and education, and good mental health, who were from the urban site. Compared to the Class 3, participants in Class 2 showed significantly higher needs, indicating that the intervention was not effective in this vulnerable group.

## Discussion

Digital navigation tools might support families that experience lower psychosocial adversity but are insufficient for families that experience higher levels of adversity, highlighting the need for tiered approaches to ensure equity.

## Introduction

Unmet social care needs, such as financial stress, housing instability, food insecurity, and limited access to social support are critical determinants of health that impact child health and development as well as family wellbeing [1]. Unmet needs can adversely impact child and family outcomes with consequent poorer health trajectories and long-term intergenerational disadvantages; this highlights the importance of early identification and intervention [2]. While several services exist to support families with these needs, access and engagement remain inequitable, especially for families from priority population communities facing multiple and intersecting challenges [3]. Health systems often lack systematic processes to identify and respond to social risks within routine care pathways. Additionally, many families are unaware of the services available to them or struggle to navigate fragmented service systems, compounding their unmet needs [4].

There is growing interest in the use of digital service navigation tools to help families report needs and receive tailored recommendations or referral support [5,6].

These tools offer the promise of scalable, efficient, and potentially empowering models of care coordination, particularly in resource-constrained environments [5,6]. However, emerging evidence suggests that the success of such tools might vary significantly depending on who uses them and when. Socioeconomic status (SES), cultural and linguistic diversity (CALD), digital literacy, and parental mental health status can all influence whether families engage meaningfully with digital navigation and whether their needs are met [7].

Watch Me Grow-Electronic (WMG-E) is a digital screening and navigation program that was developed to identify and support child developmental concerns alongside parental mental health and unmet social care needs among families with children aged 0–5 years [8,9]. Previous evaluations of the WMG-E tool have demonstrated feasibility and acceptability among families from priority populations [8,10,11]. However, less is known about how different subgroups of families within these populations respond to the intervention. In particular, it remains unclear whether certain key sociodemographic characteristics (such as child and maternal age, child's gender, parental education level, and CALD background) and clinical profiles (including parental mental health and child developmental vulnerabilities) influence engagement with the WMG-E tool or the extent to which unmet social care needs are identified and supported via the use of the tool. Furthermore, there is a limited understanding of how unmet social needs, such as housing instability, unemployment, financial stress, transportation difficulties, and food insecurity, interact with these factors to shape differential responses to digital screening and navigation support.

To address the knowledge gap, this study aims to identify the distinct subgroups of families and the underlying characteristics that influence their differential responses, if any, to the digital screening and service navigation program. By uncovering the specific contextual, demographic, and psychosocial factors associated with differential engagement with the WMG-E tool and variation in the identification and support of unmet social care needs, this study provides insights into the mechanisms that contribute to the effectiveness of the intervention across diverse populations. Findings of this research will also inform the design and implementation of targeted and personalised programs, ensuring they are effective and equitable, especially for families that are underserved or at higher risk of disengagement.

## Methods

### Study design and participants

This study is a secondary analysis of the WMG-E study, a randomised controlled trial (RCT) to evaluate the effectiveness of a digital screening and service navigation model to identify and address child developmental, parental mental health, and social care needs among two priority population communities: (i) urban and predominantly CALD communities in the South Western Sydney Local Health District (SWSLHD); and (ii) regional/rural communities of the Murrumbidgee Local Health District (MLHD). The authors confirm that all ongoing and related trials for this intervention are registered (ACTRN12621000766819).

Participants included parents/carers of a child aged between 6 months and 3 years at the time of enrollment and were recruited via opportunistic contacts (e.g., referrals from child and family health nurses (CFHNs), parenting support programs, and community playgroups, etc.). The recruitment and follow-up period of this study was between 1 August 2021 and 30 June 2023 with the recruitment commencing at the same time across both study sites. Further information on the randomisation and blinding, sample size calculation, and the intervention components are detailed in the published protocol [9]. Study findings are reported following the Strengthening the Reporting of Observational Studies in Epidemiology (STROBE) guidelines.

### Measures

#### Latent class indicators

***Parental psychological distress*** was measured using the Kessler psychological distress scale (K10) [12], a standardised questionnaire that assesses psychological distress over the previous 4 weeks. For this study, we used the four

categories of the K10 based on the total scores, including likely well (10–19), likely to have a mild mental distress (20–24), likely to have a moderate mental distress (25–29), and likely to have a severe mental distress (30–50).

**Child developmental vulnerabilities** were measured using the Learn the Signs Act Early (LTSAE) [13], a milestone-checklist used to check children's social, emotional, language, cognitive, and physical development For this study, we categorised child developmental concerns as no concerns (LTSAE = 0) and one or more concerns (LTSAE ≥1).

**Sociodemographic variables** included child's age (≤12 months, 13–24 months, ≥ 25 months), child's gender (male, female), premature birth (no, yes), maternal age (≤30 years, ≥ 31 years), marital status (married, de facto/single/divorced), parental educational level (high school or below, vocational education training, bachelor's degree and above), and socioeconomic status (SES) based on the index of relative socio-economic advantage and disadvantage (IRSAD) quintiles (quintile 1 – most disadvantaged to quintile 5 – most advantaged), and current service use (no, yes).

## Outcome variable

Unmet need was assessed using the WE CARE questionnaire [14], which screens for ten key family psychosocial issues, namely: lack of high school education, unemployment, smoking, drug abuse, alcohol abuse, depression, intimate partner violence, childcare needs, homelessness, and food insecurity. Although unmet needs was a secondary outcome of the RCT, it was used as the outcome variable and to assess associations with latent class membership following the identification of the latent classes for this study.

## Data analysis

The baseline characteristics of the sample were summarised using descriptive statistics. Categorical variables were reported as frequencies and percentages, whereas continuous variables were presented as means and standard deviations. All data were complete except for the 'current service use' variable, which had three missing cases. Given the minimal extent of missing data, multiple imputation was not undertaken, and analyses were conducted using available-case data.

Latent class analysis (LCA) was conducted to identify underlying subgroups of parents based on their sociodemographic and psychosocial characteristics. Models specifying two to five classes were estimated, and model fit was assessed using the Akaike Information Criterion (AIC), Bayesian Information Criterion (BIC), entropy values, and the Lo-Mendell-Rubin adjusted likelihood ratio test (LMR-LRT). The final model was selected based on prioritising lower AIC and BIC values, significant Lo-Mendell-Rubin likelihood ratio tests, and entropy values ≥0.70. Interpretability was assessed by examining whether classes represented conceptually meaningful and distinct clinical and sociodemographic profiles. Sensitivity analyses involved reviewing other class solutions to ensure the selected model remained robust across specifications.

Following class assignment, a zero-inflated negative binomial regression (ZINB) model with clustering to account for repeated measures at the individual level, was used to assess changes in social care needs (measured by WE CARE scores) across three time-points (baseline, 6 months, and 12 months). Clustering was handled using random intercepts for individuals to account for within-person correlations over time. Fixed effects included latent class membership, treatment group (intervention vs control), time. Interaction terms included all relevant two-way interactions (class × group, class × time, group × time) as well as the three-way interaction (class × group × time) to examine whether intervention effects differed across latent classes over time. The ZINB model was selected because WE CARE scores are discrete count data characterised by overdispersion and high proportion of zero values (approximately 50%). All analyses were conducted using R Studio (Version 2023.12.1 + 402) and Stata v19.

**Human ethics and consent to participate declarations**

The study conforms to the principles outlined in the Declaration of Helsinki. All methods were carried out in accordance with relevant guidelines and regulations of The National Statement on Ethical Conduct in Human Research (2023). The South Western Sydney Local Health District Human Research Ethics Committee approved this study (2020/ETH01418). All participating parents have provided written informed consent prior to participation.

## Results

This study recruited 288 participants with 139 in the control group and 145 in the intervention group (see **Table 1**).

### Latent class analysis

Fit statistics for 2- to 5-class models indicated that the 3-class model provided the best balance between fit and parsimony (AIC = 4243.0; BIC = 4436.4; entropy = 0.73), with significant improvement over the 2-class model based on the LMR-LRT ($p < 0.001$) (see **Table 2**). The 3-class solution was retained for interpretation.

**Class 1 (non-CALD, married, and educated; $n = 134$)** primarily included parents older than 30 years of age (61.7%), with high levels of education (72.6% holding postgraduate qualifications), who were predominantly married (80.7% married), middle-class (54.5% in Quintile 3 IRSAD), from the MLHD rural/regional site (89.0%), who were entirely non-CALD population (100.0%), and generally reported good mental health (with 65.7% likely to be well).

**Class 2 (younger parents, non-CALD, low education and SES with poor mental health; $n = 94$)** was comprised of younger parents (70.1% under 30 years of age) with lower education levels (52.3% Year 12 or below), who were predominantly unmarried (82.3% de facto, single, or divorced), almost entirely non-CALD (97.5%), with a lower SES with a notable presence in the most disadvantaged quintiles (27.5% in Quintile 1) from the MLHD rural/regional site (90.0%), and with poor maternal mental health (with 19.6% likely to have severe mental distress).

**Class 3 (older parents, CALD, high SES, married, highly educated, very good mental health; $n = 56$)** included older parents (80.5% over 30 years), who were highly educated (75.9% postgraduate qualifications), with a significant proportion from CALD backgrounds (67.9%), who were married (90.5%), of higher SES (17.0% in Quintiles 4–5) from the SWS urban site (66.0%), with few child developmental concerns (84.9% showing no issues), and with mostly positive mental health (with 74.6% likely to be well; see **Table 3**).

### Trajectories of social care needs

Findings of the mixed-effects models showed that participants in Class 2 had significantly higher social care needs compared to Class 3 (reference group), indicating that this subgroup experienced greater vulnerability. No other significant main effects or interactions were observed between Class memberships and treatment groups (intervention vs. control) over time in (see **Fig 1** and **Table 4**). While not statistically significant, there were several notable trends. Compared to Class 3 control participants, participants in the intervention group of class 1 showed a trend towards improvement over time whereas those in Class 2 showed increased needs over time, indicating that the intervention was not effective in this vulnerable group.

## Discussion

### Summary of findings

This study identified three distinct latent subgroups of parents based on their sociodemographic and psychosocial characteristics, which were associated with differences in social care needs and responses to the WMG-E digital screening and navigation intervention. Findings showed that Class 2 represented a more vulnerable subgroup, comprised primarily of younger, less-educated, non-CALD parents who were unmarried and predominantly from rural/regional site, with greater

**Table 1. Descriptive characteristics of the sample (N = 288).**

| Characteristics | Control (n = 139) | Intervention (n = 145) |
|---|---|---|
| Child's age | | |
| ≤12 months | 44 (31.7%) | 50 (34.5%) |
| 13 to 24 months | 49 (35.3%) | 55 (37.9%) |
| ≥25 months | 46 (33.1%) | 40 (27.6%) |
| Child's gender | | |
| Male | 67 (48.2%) | 74 (51.0%) |
| Female | 72 (51.8%) | 71 (49.0%) |
| Premature birth | | |
| No | 132 (95.0%) | 138 (95.2%) |
| Yes | 7 (5.0%) | 7 (4.8%) |
| Child's developmental concern (LTSAE) | | |
| No concern | 113 (81.3%) | 119 (82.1%) |
| One or more concerns | 26 (18.7%) | 26 (17.9%) |
| Maternal age | | |
| ≤30 years | 62 (44.6%) | 64 (44.1%) |
| ≥31 years | 77 (55.4%) | 81 (55.9%) |
| Marital status | | |
| Married | 91 (65.5%) | 85 (58.6%) |
| De facto, Single, or Divorced | 48 (34.5%) | 60 (41.4%) |
| CALD status | | |
| No | 115 (82.7%) | 119 (82.1%) |
| Yes | 24 (17.3%) | 26 (17.9%) |
| Highest education level | | |
| High school or below | 23 (16.5%) | 38 (26.2%) |
| Vocational education training | 38 (27.3%) | 38 (26.2%) |
| Bachelor's degree and above | 78 (56.1%) | 69 (47.6%) |
| Socioeconomic status (IRSAD) | | |
| Quintile 1 (most disadvantaged) | 24 (17.3%) | 33 (22.8%) |
| Quintile 2 | 57 (41.0%) | 43 (29.7%) |
| Quintile 3 | 47 (33.8%) | 61 (42.1%) |
| Quintile 4 | 7 (5.0%) | 4 (2.8%) |
| Quintile 5 (most advantaged) | 4 (2.9%) | 4 (2.8%) |
| Site | | |
| South West Sydney | 30 (21.6%) | 31 (21.4%) |
| Murrumbidgee | 109 (78.4%) | 114 (78.6%) |
| Current service uptake | | |
| No | 105 (76.1%) | 103 (72.5%) |
| Yes | 33 (23.9%) | 39 (27.5%) |
| Parental mental health (K10) | | |
| Likely to be well | 85 (61.2%) | 95 (65.5%) |
| Likely to have a mild mental disorder | 28 (20.1%) | 21 (14.5%) |
| Likely to have a moderate mental disorder | 13 (9.4%) | 16 (11.0%) |
| Likely to have a severe mental disorder | 13 (9.4%) | 13 (9.0%) |
| Mean social care needs score (WE CARE) | | |

*(Continued)*

**Table 1.** (Continued)

| Characteristics | Control (n = 139) | Intervention (n = 145) |
|---|---|---|
| *Baseline* | 1.32 (1.5) | 1.10 (1.5) |
| *6-months* | 1.18 (1.5) | 1.14 (1.8) |
| *12-months* | 1.02 (1.3) | 0.95 (1.6) |

LTSAE – Learn the Signs Act Early; CALD – culturally and linguistically diverse; IRSAD – index of relative socio-economic advantage and disadvantage; K10 – Kessler psychological distress scale.

**Table 2. Goodness of fit statistics for 2 to 5-class solutions.**

| Classes | AIC[d] | BIC[e] | Lo-Mendell- Rubin LR test | Entropy |
|---|---|---|---|---|
| 2 | 4268.3 | 4396.0 | <0.001 | 0.66 |
| **3** | **4243.0** | **4436.4** | **<0.001** | **0.73** |
| 4 | 4231.8 | 4490.9 | <0.001 | 0.73 |
| 5 | 4237.1 | 4561.8 | <0.001 | 0.80 |

[d]Akaike information criteria; [e]Bayesian information criteria; LR – likelihood ratio

socioeconomic disadvantage, including elevated levels of psychological distress, who reported significantly higher unmet needs compared to other classes. Additionally, there were some notable trends in the effectiveness of the intervention that appeared to vary across classes, with vulnerable families in Class 2 facing persistent needs over time. These findings underscore the marked heterogeneity based on family circumstances and emphasise that a one-size-fits-all approach might not effectively address complex, intersecting needs across diverse population groups. This also highlights the critical need for tailored interventions to better support the challenges and capacities of different family groups.

## Impact of the intervention by the class membership

Analysis of social care needs over time showed that Class 1 participants in the intervention group demonstrated a non-significant but consistent trend of improvement, suggesting greater capacity to engage and act on the digital intervention. Class 3 had low baseline needs and this was maintained over time, consistent with their strong socioeconomic profile, psychosocial supports, and wellbeing. In contrast, Class 2 – representing the most disadvantaged group – reported higher unmet social care needs and a persistent trend of increasing needs post-intervention, indicating that more intensive support might be required beyond the digital intervention alone.

While the findings suggest that digital screening and navigation tools, such as WMG-E, can identify and support families with less complex needs, families with multiple, intersecting vulnerabilities might need more intense support, such as face-to-face navigation, co-located services, such as integrated child and family hubs [15,16]. Specifically, digital navigation tools, while scalable and resource-efficient, might not be suitable to address barriers faced by parents with limited digital literacy, social capital, or psychological distress. Hence, it is critical to design and implement interventions within a 'proportionate universalism framework' characterised by universal care plus targeted supports commensurate with needs, in a tiered care model with the level and type of support based on the unique child and family-level profiles [17].

The differential class-specific findings from this study reflect patterns in previous research on digital health tools and service navigation. Research has shown that universal digital interventions often benefit higher SES and digitally literate populations, more than people with complex needs and/or lower digital access [18–20]. The finding that Class 1 participants showed the most consistent but non-significant improvement aligns with the inverse care law [21], as families with

**Table 3. Predicted probabilities of class membership for the three-class model.**

| Characteristics | Class 1 (n = 134) | Class 2 (n = 94) | Class 3 (n = 56) |
|---|---|---|---|
| Child's age | | | |
| ≤12 months | 0.3375 | 0.3838 | 0.2484 |
| 13 to 24 months | 0.3272 | 0.3468 | 0.4577 |
| ≥25 months | 0.3353 | 0.2694 | 0.2939 |
| Child's gender | | | |
| Male | 0.5111 | 0.5545 | 0.3932 |
| Female | 0.4889 | 0.4455 | 0.6068 |
| Premature birth | | | |
| No | 0.9230 | 0.9485 | 1.0000 |
| Yes | 0.0770 | 0.0515 | 0.0000 |
| Child's developmental concern (LTSAE) | | | |
| No concern | 0.8312 | 0.7755 | 0.8492 |
| One or more concerns | 0.1688 | 0.2245 | 0.1508 |
| Maternal age | | | |
| ≤30 years | 0.3833 | 0.7010 | 0.1950 |
| ≥31 years | 0.6167 | 0.2990 | 0.8050 |
| Marital status | | | |
| Married | 0.8068 | 0.1797 | 0.9048 |
| De facto, Single, or Divorced | 0.1932 | 0.8203 | 0.0952 |
| CALD status | | | |
| No | 1.0000 | 0.9785 | 0.3201 |
| Yes | 0.0000 | 0.0215 | 0.6799 |
| Highest education level | | | |
| High school or below | 0.0000 | 0.5231 | 0.1549 |
| Advanced Diploma or Grad Cert 3/4 | 0.2742 | 0.3939 | 0.0852 |
| Graduate Diploma or PG degree | 0.7258 | 0.0831 | 0.7599 |
| Socioeconomic status (IRSAD) | | | |
| Quintile 1 (most disadvantaged) | 0.1020 | 0.2747 | 0.2651 |
| Quintile 2 | 0.2957 | 0.3468 | 0.4536 |
| Quintile 3 | 0.5448 | 0.3785 | 0.1079 |
| Quintile 4 | 0.0142 | 0.0000 | 0.1322 |
| Quintile 5 (most advantaged) | 0.0433 | 0.0000 | 0.0411 |
| Site | | | |
| South West Sydney | 0.1120 | 0.0960 | 0.6610 |
| Murrumbidgee | 0.8880 | 0.9040 | 0.3390 |
| Current service uptake | | | |
| No | 0.7650 | 0.6740 | 0.8040 |
| Yes | 0.2350 | 0.3260 | 0.1960 |
| Parental mental health (K10) | | | |
| Likely to be well | 0.6570 | 0.5229 | 0.7456 |
| Likely to have a mild mental disorder | 0.2060 | 0.1455 | 0.1534 |
| Likely to have a moderate mental disorder | 0.1259 | 0.1358 | 0.0166 |
| Likely to have a severe mental disorder | 0.0111 | 0.1958 | 0.0844 |

LTSAE – Learn the Signs Act Early; CALD – culturally and linguistically diverse; IRSAD – index of relative socio-economic advantage and disadvantage; K10 – Kessler psychological distress scale.

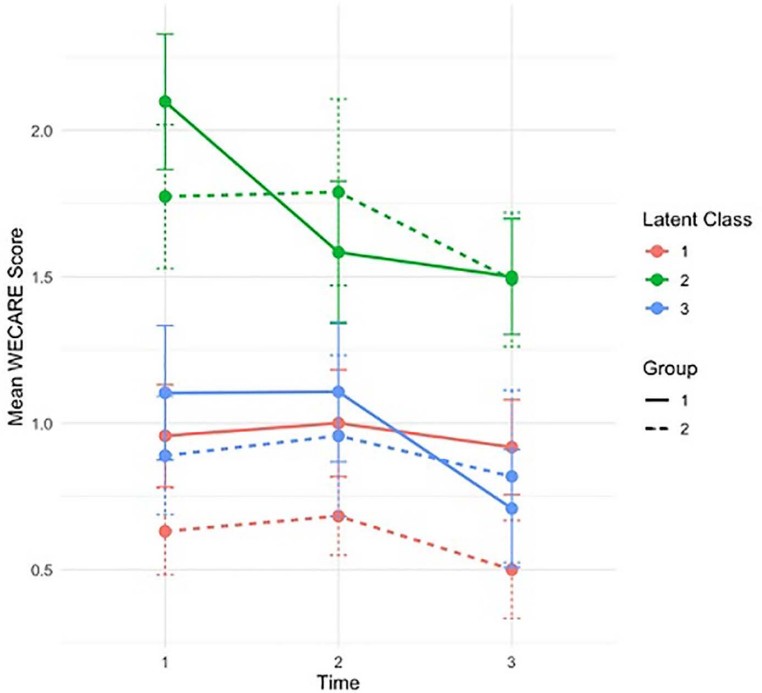

**Fig 1. Trajectories of social care needs over time by class memberships and group.**

lower needs were more likely to have the resources, time, and confidence to benefit from digital navigation and act on service recommendations. In contrast, the negative trend observed in Class 2 resonates with research that has highlighted the limitations of digital-only solutions in high-need populations. For example, research in primary care and maternal-child health contexts has shown that families facing cumulative stressors, such as young parenthood, financial hardship, and mental health challenges, require intensive, personalised navigation, or case management, rather than self-guided digital tools alone [22–24]. While digital and virtual approaches are attractive from a scalability perspective, they might increase inequitable outcomes if they are implemented as the sole model of care.

The increase in reported social care needs among Class 2 participants likely reflects the identification of unmet needs without sufficient capacity or intensity of intervention to address them, a phenomenon highlighted in prior work examining social risk screening in healthcare settings [25]. In such contexts, screening without effective, accessible, and timely intervention pathways may unintentionally exacerbate inequities, particularly for families facing structural and systemic barriers to care, including limited financial resources, reduced local service availability, transportation challenges, and difficulties balancing work or caregiving responsibilities. These systemic barriers are particularly salient in rural or regional settings, where services are often less accessible, waitlists are longer, and specialist supports may be scarce. This interpretation is supported by the qualitative findings from the broader trial, where families reported challenges in following through with digital referrals without hands-on support [26]. Moreover, the study was conducted during the COVID-19 pandemic associated social restrictions and health service shutdowns likely exacerbated difficulties in accessing the recommended supports. For families with low social capital and limited digital literacy, these factors may have further hindered their ability to engage with the intervention and receive timely support. Finally, this study adds to the emerging literature advocating for stratified or tiered care models, where digital tools are matched with the appropriate level of personal support using hybrid care models based on complexity and risk [27–29]. Tailoring the intensity and modality of interventions to user profiles can enhance both efficiency and equity, particularly in early childhood and family support settings.

**Table 4. Trajectories of social care needs by treatment condition, time, and latent class membership interactions.**

| Effect | Estimate | 95%CI | p-value |
|---|---|---|---|
| Intercept | 0.20 | −0.25, 0.66 | 0.378 |
| Latent Class membership | | | |
| Class 1 | −0.08 | −0.62, 0.46 | 0.778 |
| Class 2 | 0.58 | 0.04, 1.13 | **0.035** |
| Class 3 | Reference | | |
| Group | | | |
| Control | Reference | | |
| Intervention | −0.24 | −0.90, 0.42 | 0.477 |
| Time | | | |
| Baseline | Reference | | |
| 6 months | 0.01 | −0.63, 0.64 | 0.984 |
| 12 months | −0.40 | −1.13, 0.32 | 0.275 |
| Class x Group | | | |
| Class 3 x Control | Reference | | |
| Class 1 x Intervention | −0.13 | −0.95, 0.69 | 0.755 |
| Class 2 x Intervention | 0.15 | −0.64, 0.93 | 0.713 |
| Class x Time | | | |
| Class 3 x Baseline | Reference | | |
| Class 1 x 6 months | 0.05 | −0.72, 0.82 | 0.895 |
| Class 1 x 12 months | 0.37 | −0.48, 1.23 | 0.394 |
| Class 2 x 6 months | −0.23 | −1.02, 0.56 | 0.573 |
| Class 2 x 12 months | 0.13 | −0.76, 1.02 | 0.777 |
| Time x Group | | | |
| Baseline x Control | Reference | | |
| 6 months x Intervention | 0.10 | −0.86, 1.06 | 0.835 |
| 12 months x Intervention | 0.35 | −0.69, 1.39 | 0.508 |
| Class x Group x Time | | | |
| Class 3 x Control x Baseline | Reference | | |
| Class 1 x Intervention x 6 months | −0.09 | −1.27, 1.08 | 0.878 |
| Class 1 x Intervention x 12 months | −0.57 | −1.84, 0.70 | 0.378 |
| Class 2 x Intervention x 6 months | 0.11 | −1.04, 1.25 | 0.855 |
| Class 2 x Intervention x 12 months | −0.26 | −1.50, 0.97 | 0.376 |

Adjusted for covariates.

The findings also raise important considerations around cultural safety and digital inclusion. Class 3 (predominantly CALD families) showed lower social care needs and better mental health profile. While this might be a function of better education and hence better health literacy and capacity to access supports, it remains possible that these families under-reported needs due to cultural stigma, limited trust in health systems, or language barriers. Further, while the WMG-E had language translations available for screening, interpreter availability might enhance accessibility and uptake across culturally diverse populations.

## Implications for equity and service design

The findings of this study suggest that a 'one-size-fits-all' intervention might be inappropriate to address social care needs in diverse priority populations. These findings highlight the need for stratified or tiered models of digital

care navigation, in which the intensity of support is tailored to participant profiles. For families with fewer barriers to service engagement, digital tools might be sufficient to prompt action and connect them to relevant supports. For families that experience greater disadvantage, however, digital tools must be embedded within more intensive, relationship-based approaches, such as in-person, community health worker programs, case management, and/or co-location of social support services within place-based settings, such as integrated child and family hubs [30]. Additionally, the results highlight the potential of latent class analysis to inform stratification in digital health intervention models. Designing such models requires explicit attention to the social determinants of health, particularly access, inequity, and socioeconomic disadvantage as well as contextual factors including rurality and the role of social stigma in some CALD communities, which can affect both engagement with services and resulting outcomes. Integrating brief screening tools at the point of enrolment might allow programs to dynamically allocate families to different levels of navigation support based on the family's needs, preferences, and capacity to engage with services.

### Strengths and limitations

A key strength of this study is the use of LCA to uncover heterogeneity in the sample and evaluate differential intervention effects on psychosocial risk factors. Additionally, the use of longitudinal data also allowed for an assessment of change over time, within and between subgroups. Nevertheless, this study has some limitations. First, the class membership was probabilistic and based on observed indicators collected as part of the study, which might not capture all relevant aspects of family complexity or underlying causal relationships. Second, although the sample was drawn from a real-world service setting, the sample size within certain latent classes was relatively small, potentially limiting statistical power for subgroup analyses. Third, social care needs were self-reported and might be subject to response bias or changes in perception over time, particularly after exposure to a needs-assessment tool.

### Future directions

Future studies should explore adaptive intervention designs that modify navigation intensity based on participant characteristics or early engagement patterns. Targeted sampling to include diverse priority population groups is also essential to ensure findings are relevant and equitable. Mixed-methods research – particularly qualitative approaches and arts-based research methods, can illuminate the lived experiences of families in each class, shedding light on barriers to acting on referrals, digital literacy challenges, and the emotional impact of navigating complex systems with minimal support. Moreover, there is a need to pilot-test and evaluate blended models that combine digital tools with human navigation, particularly for populations with higher psychosocial risk. Embedding navigators within trusted community-based or primary care settings and ensuring cultural competence might enhance trust and effectiveness [6].

### Conclusion

Digital navigation tools can reach families experiencing service access barriers in resource constrained environments, such as in regional and rural communities. While these tools might be effective for families with less complex needs and good parental capacity, digital service navigation might be insufficient to support families experiencing significant psychosocial adversity. Stratified approaches that leverage a proportionate universalism model (universal plus targeted additional support), matching navigation intensity to participant needs, are essential to promote equity and effectiveness in digital health interventions.

### Acknowledgments

We would like to express our sincere gratitude to the families who generously shared their time and experiences to participate in this study. Their contributions were invaluable to advancing our understanding of how digital screening and

navigation models can better support child development and family wellbeing. We also thank the staff and administrators from South Western Sydney Local Health District (SWSLHD) and Murrumbidgee Local Health District (MLHD) for their support in facilitating recruitment and implementation at each site. In particular, we acknowledge the contributions of the Child and Family Health Services teams, local research governance officers, and service navigator staff who played a vital role in supporting participants throughout the study. Finally, we thank the broader project team and support staff whose efforts in data management, project coordination, and community engagement made this work possible but who are not listed as co-authors.

## Author contributions

**Conceptualization:** James R John, John Preddy, Susan Woolfenden, Valsamma Eapen.

**Data curation:** James R John, Teresa Winata, Si Wang, Melissa Smead, Weng Tong Wu, Christa Lam-Cassettari.

**Formal analysis:** James R John.

**Funding acquisition:** Jane Kohlhoff, Virginia Schmied, Bin Jalaludin, Kenny Lawson, Siaw-Teng Liaw, Raghu Lingam, Andrew Page, Katherine Boydell, Ping-I Lin, Ilan Katz, Ann Dadich, Shanti Raman, Rebekah Grace, Aunty Kerrie Doyle, Tom McClean, John Preddy, Susan Woolfenden, Valsamma Eapen.

**Investigation:** Si Wang, Melissa Smead, Weng Tong Wu, Valsamma Eapen.

**Methodology:** James R John, Teresa Winata, Jane Kohlhoff, Virginia Schmied, Bin Jalaludin, Kenny Lawson, Siaw-Teng Liaw, Raghu Lingam, Andrew Page, Christa Lam-Cassettari, Katherine Boydell, Ping-I Lin, Ilan Katz, Ann Dadich, Shanti Raman, Rebekah Grace, Aunty Kerrie Doyle, Blaise Di Mento, John Preddy, Susan Woolfenden, Valsamma Eapen.

**Supervision:** John Preddy, Susan Woolfenden, Valsamma Eapen.

**Writing – original draft:** James R John, Teresa Winata.

**Writing – review & editing:** James R John, Teresa Winata, Si Wang, Melissa Smead, Weng Tong Wu, Jane Kohlhoff, Virginia Schmied, Bin Jalaludin, Kenny Lawson, Siaw-Teng Liaw, Raghu Lingam, Andrew Page, Christa Lam-Cassettari, Katherine Boydell, Ping-I Lin, Ilan Katz, Ann Dadich, Shanti Raman, Rebekah Grace, Aunty Kerrie Doyle, Tom McClean, Blaise Di Mento, John Preddy, Susan Woolfenden, Valsamma Eapen.

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
