## [Decision Letter · Decision Letter 0]

19 Nov 2025

Dear Dr. Eapen,

Thank you for submitting your manuscript to PLOS ONE. After careful consideration, we feel that it has merit but does not fully meet PLOS ONE’s publication criteria as it currently stands. Therefore, we invite you to submit a revised version of the manuscript that addresses the points raised during the review process.

We look forward to receiving your revised manuscript.

Kind regards,

Onaedo Ilozumba

Academic Editor

PLOS ONE

2. In the online submission form, you indicated that [The datasets generated and/or analysed during the current study are not publicly available due to ethical and legal reasons, including participant anonymity and privacy, but may be available from the corresponding author on reasonable request.].

Additional Editor Comments (if provided):

Reviewers' comments:

Reviewer's Responses to Questions

**Comments to the Author**

1. Is the manuscript technically sound, and do the data support the conclusions?

Reviewer #1: Yes

Reviewer #2: Yes

2. Has the statistical analysis been performed appropriately and rigorously?

Reviewer #1: I Don't Know

Reviewer #2: Yes

3. Have the authors made all data underlying the findings in their manuscript fully available?

Reviewer #1: Yes

Reviewer #2: Yes

4. Is the manuscript presented in an intelligible fashion and written in standard English?

Reviewer #1: Yes

Reviewer #2: Yes

Reviewer #1: Overall: The manuscript provides an analysis of data collected in context of larger RCT in which they use LCA methods to identify characteristics associated with social risks and with response to digital intervention. Rural, poor, mentally ill, non-CALD populations were more likely to have social risks (by definition, since they're poor and otherwise marginalized) and trended towards being less responsive to digital intervention. No big surprises there. I would suggest authors do a little revising to abstract to clarify what they are doing, strengthen intro to contextualize work, and add to Discussion to think more deeply about what it means that largely poor, rural, mentally ill, but NONCALD people seemed to do worse. They don't really engage with the finding around race/language and in fact seem to suggest later in discussion that more linguistic support is necessary, which feels inconsistent with findings. Additional notes that I took while reading are included below.

Abstract: Opening sentence is: "Digital tools are increasingly integrated into health systems to identify and support needs, yet their effectiveness remains unclear among priority communities.” The relationship between digital tools and needs is unclear. Whose needs? Effectiveness for what? Second sentence transitions to “digital intervention” but what relationship does that have with digital tools? Just needs clearer link to what they’re studying in this article. Somewhere in abstract it would be helpful to explain that authors are looking at patient, caregiver/household needs. For instance, in methods section of abstract, it would help orient reader if th word “patient” or “household” were inserted before “clinical and sociodemographic” in this sentence: "Latent class analysis was used to identify subgroups based on clinical and sociodemographic factors."

Introduction: They indicate intervention has been evaluated in RCT, but don’t clearly share findings from the evaluation unless that’s the references 8 and 10? The structure of the first five sentences of that paragraph is really confusing to a naive reader. Then the next half of that paragraph is about what is unknown. Do the authors really mean to say that it is unclear in literature how parental education is related to unmet needs?

Discussion: Opening of summary of findings suggests that class shaped needs and response to intervention. I don’t think that is accurate. Class was associated with needs; using language suggesting causality here is not quite right. At end of that section, they suggest, "This also highlights the critical need for tailored interventions, particularly digital models of care, to better support the challenges and capacities of different family groups.” Once I read rest of discussion, I understood (and agreed) with their point, but this sentence is a little ambiguous. In fact, because they did not test digital models vs other model in this study, they don’t know if non digital models “particularly” need tailoring. Also wondering if authors can explain why they are suggesting more linguistic support (they refer to language barriers, e.g.) necessary when Class 2 is largely nonCALD. I agree with suggestion, but not c/w findings, right? Or maybe I'm misunderstanding. Also maybe could describe a little more about rural/urban context in Australia and how they make sense of that.

Reviewer #2: As the statistical reviewer I will focus on methods and reporting

Major

1) how did you deal with missing data? were all data complete? please state so if yes. if not, why weren't multiple imputation approaches considered?

2) the authors said "final model was selected based on the optimal balance of fit indices and interpretability". perhaps some more information is needed here for clarity, and to make analyses replicable. Specify which fit indices were prioritised (AIC, LMR-LRT other), and how interpretability was assessed. Was any sensitivity analysis conducted to assess robustness of the chosen model?"

Minor

1) what is regression for count data? Poisson? negative binomial regression? can the authors be clearer here? made clearer later ,but you don't have to report things twice, the first time being unclear

2) "...and their interactions". can you be clearer on the interactions included in the model? it becomes a bit clearer later i think, trial arm by latent class

3) How was clustering accounted for? using random effects or robust standard errors? clarify how repeated measures were handled as well.

**Do you want your identity to be public for this peer review?** For information about this choice, including consent withdrawal, please see our Privacy Policy

Reviewer #1: No

Reviewer #2: No

---

## [Author Response · Author response to Decision Letter 1]

1 Dec 2025

Point-by-point response to Reviewers’ comments

Dear Editor,

We thank you and the reviewers for your time in reviewing our paper and providing valuable comments that have helped to improve the current version. We have carefully considered the comments and responded to all the items. We hope the manuscript after careful revisions meet your high standards. Below we provide the point-by-point responses. All modifications in the manuscript have been highlighted as tracked changes.

Author response: We have carefully formatted the manuscript as per PLOS ONE's style requirements.

2. In the online submission form, you indicated that [The datasets generated and/or analysed during the current study are not publicly available due to ethical and legal reasons, including participant anonymity and privacy, but may be available from the corresponding author on reasonable request.].

Author response: We have revised the data sharing statement in the manuscript (See Page 25, Lines 390-396).

Modifications: “Data sharing statement: The individual level data collected and analysed during the current study are not publicly available due to ethical and privacy reasons. Data sharing has been restricted by the South Western Sydney Local Health District Human Research Ethics Committee. Requests will be reviewed in accordance with the approval of the South Western Sydney Local Health District Human Research Ethics Committee (swslhd-ethics@health.nsw.gov.au) to ensure that all ethical and legal requirements for data protection are met.”

Author response: We confirm that the ethics statement only appears in the Methods section of your manuscript.

Author response: We thank the Editorial team for this information.

Reviewer #1

1. Overall: The manuscript provides an analysis of data collected in context of larger RCT in which they use LCA methods to identify characteristics associated with social risks and with response to digital intervention. Rural, poor, mentally ill, non-CALD populations were more likely to have social risks (by definition, since they're poor and otherwise marginalized) and trended towards being less responsive to digital intervention. No big surprises there. I would suggest authors do a little revising to abstract to clarify what they are doing, strengthen intro to contextualize work, and add to Discussion to think more deeply about what it means that largely poor, rural, mentally ill, but NONCALD people seemed to do worse. They don't really engage with the finding around race/language and in fact seem to suggest later in discussion that more linguistic support is necessary, which feels inconsistent with findings. Additional notes that I took while reading are included below.

Author response: We thank the reviewer for this feedback. We have revised the manuscript in several key areas. First, we have clarified the study aims in the Abstract to better reflect our analytical focus and intended contribution (See Page 3, Lines 49-52). Second, we have strengthened the Introduction to provide clearer context around the populations most affected by unmet social care needs and the importance of examining differential responses to digital service navigation (See Page 6, Lines 102-112). Third, we have expanded the Discussion to engage more deeply with the finding that participants who were predominantly from low socioeconomic status, rural, and experiencing mental health difficulties but not from CALD backgrounds had poorer outcomes (See Pages 20-21, Lines 312-327).

Abstract

1. Opening sentence is: "Digital tools are increasingly integrated into health systems to identify and support needs, yet their effectiveness remains unclear among priority communities.” The relationship between digital tools and needs is unclear. Whose needs? Effectiveness for what?

Author response: We agree that the opening sentence in the abstract is ambiguous. We have revised the text to explicitly state that we are referring to the unmet social care needs of families and the effectiveness of these tools in addressing those needs (See Page 3, Lines 49-51).

Modifications: “Digital screening and navigation interventions are increasingly integrated into health systems to identify and support families’ unmet social care needs, yet their effectiveness in improving outcomes remains unclear among priority communities.”

2. Second sentence transitions to “digital intervention” but what relationship does that have with digital tools? Just needs clearer link to what they’re studying in this article.

Author response: We have now refined the first sentence to start with “Digital screening and navigation interventions” to better link with the second sentence and ensure consistency (See Page 3, Lines 49-52).

Modifications: “Digital screening and navigation interventions are increasingly integrated into health systems to identify and support families’ unmet social care needs, yet their effectiveness in improving outcomes remains unclear among priority communities. We hypothesise that responses to such digital interventions might vary based on sociodemographic and psychosocial characteristics.”

3. Somewhere in abstract it would be helpful to explain that authors are looking at patient, caregiver/household needs. For instance, in methods section of abstract, it would help orient reader if the word “patient” or “household” were inserted before “clinical and sociodemographic” in this sentence: "Latent class analysis was used to identify subgroups based on clinical and sociodemographic factors."

Author response: We have revised the sentence and added “subgroup of families” and based on “parental and child” clinical and sociodemographic factors for clarity (See Page 3, Lines 56-57).

Modifications: “Latent class analysis was used to identify subgroups of families based on parental and child clinical and sociodemographic factors.”

Introduction

1. They indicate intervention has been evaluated in RCT, but don’t clearly share findings from the evaluation unless that’s the references 8 and 10? The structure of the first five sentences of that paragraph is really confusing to a naive reader.

Author response: We thank the reviewer for this feedback. We have rewritten the first five sentences of the paragraph in the introduction to clarify the findings from previous evaluations of the WMG-E tool. Specifically, we now explicitly state that prior evaluations have demonstrated the feasibility, acceptability, and ability of the WMG-E tool to increase service connections among families from priority populations (See Pages 5-6, Lines 93-102).

Modifications: “Watch Me Grow-Electronic (WMG-E) is a digital screening and navigation program that was developed to identify and support child developmental concerns alongside parental mental health and unmet social care needs among families with children aged 0-5 years [8, 9]. Previous evaluations of the WMG-E tool have demonstrated feasibility and acceptability among families from priority populations [8, 10, 11].”

2. Then the next half of that paragraph is about what is unknown. Do the authors really mean to say that it is unclear in literature how parental education is related to unmet needs?

Author response: We appreciate the reviewer’s comment and have clarified this point. We do not intend to suggest that the literature is unclear about the general relationship between parental education and unmet needs. Rather, we now explicitly state that it is unclear how parental education, alongside other sociodemographic and clinical characteristics, influences families’ engagement with the WMG-E tool and the extent to which the intervention addresses unmet social care needs (See Page 6, Lines 102-112).

Modifications: “However, less is known about how different subgroups of families within these populations respond to the intervention. In particular, it remains unclear whether certain key sociodemographic characteristics (such as child and maternal age, child’s gender, parental education level, and cultural background) and clinical profiles (including parental mental health and child developmental vulnerabilities) influence engagement with the WMG-E tool or the extent to which unmet social care needs are identified and supported via the use of the tool. Furthermore, there is a limited understanding of how unmet social needs, such as housing instability, unemployment, financial stress, transportation difficulties, and food insecurity, interact with these factors to shape differential responses to digital screening and navigation support.”

Discussion

1. Opening of summary of findings suggests that class shaped needs and response to intervention. I don’t think that is accurate. Class was associated with needs; using language suggesting causality here is not quite right.

Author response: We agree that causal language is inappropriate here. We have modified the text to reflect association rather than causation (See Page 19, Lines 266-268).

Modifications: “This study identified three distinct latent subgroups of parents based on their sociodemographic and psychosocial characteristics, which were associated with differences in social care needs and responses to the WMG-E tool.”

2. At end of that section, they suggest, "This also highlights the critical need for tailored interventions, particularly digital models of care, to better support the challenges and capacities of different family groups.” Once I read rest of discussion, I understood (and agreed) with their point, but this sentence is a little ambiguous. In fact, because they did not test digital models vs other model in this study, they don’t know if non digital models “particularly” need tailoring.

Author response: We thank the reviewer for this feedback. We agree that the original wording could be misinterpreted as implying that digital models require more tailoring than other non-digital models. We have revised the sentence to remove this implication and ensure it accurately reflects the study findings without suggesting comparisons between digital and non-digital models which was not the focus of this study (See Page 19, Lines 276-277).

Modifications: “This also highlights the critical need for tailored interventions to better support the challenges and capacities of different family groups.”

3. Also wondering if authors can explain why they are suggesting more linguistic support (they refer to language barriers, e.g.) necessary when Class 2 is largely non CALD. I agree with suggestion, but not c/w findings, right? Or maybe I'm misunderstanding. Also maybe could describe a little more about rural/urban context in Australia and how they make sense of that.

Author response: We agree with the feedback. We have revised the discussion to highlight the multiple factors contributing to the vulnerabilities of Class 2 families, including socioeconomic disadvantage, rural/remote location, limited access to services, and psychological distress. This clarifies the barriers faced by this subgroup and better contextualises their persistent unmet needs (See Pages 20-21, Lines 312-327).

Modifications: “The increase in reported social care needs among Class 2 participants reflects the concept of ‘diagnostic overload’, where screening identifies issues but fails to provide effective pathways to resolution, posing a risk of increased inequity [25]. Families in this group may experience multiple, intersecting barriers such as limited financial resources, reduced local service availability, transportation challenges, and difficulties balancing work or caregiving responsibilities which can exacerbate the challenges of accessing recommended supports. These systemic barriers are particularly salient in rural or regional settings, where services are often less accessible, waitlists are longer, and specialist supports may be scarce. This interpretation is supported by the qualitative findings from the broader trial, where families reported challenges in following through with digital referrals without hands-on support [26]. Moreover, the study was conducted during the COVID-19 pandemic associated social restrictions and health service shutdowns likely exacerbated difficulties in accessing the recommended supports. For families with low social capital and limited digital literacy, these factors may have further hindered their ability to engage with the intervention and receive timely support.”

Reviewer #2

As the statistical reviewer I will focus on methods and reporting

Major

1) how did you deal with missing data? were all data complete? please state so if yes. if not, why weren't multiple imputation approaches considered?

Author response: We thank the reviewer for this comment. All data were complete except for one variable (current service use), which had only three missing cases. Given the extremely small amount of missing data, multiple imputation was not necessary, and analyses were conducted using the available data.

Modifications: No changes made.

2) the authors said "final model was selected based on the optimal balance of fit indices and interpretability". perhaps some more information is needed here for clarity, and to make analyses replicable. Specify which fit indices were prioritised (AIC, LMR-LRT other), and how interpretability was assessed. Was any sensitivity analysis conducted to assess robustness of the chosen model?"

Author response: We have now elaborated on the model selection process in the Methods section (See Page 9, Lines 176-181).

Modifications: “The final model was selected based on prioritising lower AIC and BIC values, significant Lo-Mendell-Rubin likelihood ratio tests, and entropy values ≥0.70. Interpretability was assessed by examining whether classes represented conceptually meaningful and distinct clinical and sociodemographic profiles. Sensitivity analyses involved reviewing other class solutions to ensure the selected model remained robust across specifications.”

Minor

1) what is regression for count data? Poisson? negative binomial regression? can the authors be clearer here? made clearer later, but you don't have to report things twice, the first time being unclear

Author response: We have clarified the statistical method earlier in the Data Analysis section to avoid ambiguity (See Page 9, Lines 183-184).

Modifications: “Following class assignment, a zero-inflated negative binomial regression (ZINB) model

---

## [Decision Letter · Decision Letter 1]

16 Dec 2025

Dear Dr. Eapen,

Thank you for submitting your manuscript to PLOS ONE. After careful consideration, we feel that it has merit but does not fully meet PLOS ONE’s publication criteria as it currently stands. Therefore, we invite you to submit a revised version of the manuscript that addresses the points raised during the review process.

We look forward to receiving your revised manuscript.

Kind regards,

Onaedo Ilozumba

Academic Editor

PLOS One

**Journal Requirements:**

**Additional Editor Comments:**

The reviewers appreciated your attention to their suggestions and had only a few  minor comments. Please note the request to include a statement about your missing data within the methods section of your manuscript. We would encourage you to resubmit these minor corrections as soon as you can. 

Reviewers' comments:

Reviewer's Responses to Questions

**Comments to the Author**

Reviewer #1: All comments have been addressed

Reviewer #2: All comments have been addressed

2. Is the manuscript technically sound, and do the data support the conclusions?

Reviewer #1: Yes

Reviewer #2: Yes

3. Has the statistical analysis been performed appropriately and rigorously?

Reviewer #1: I Don't Know

Reviewer #2: Yes

4. Have the authors made all data underlying the findings in their manuscript fully available?

Reviewer #1: Yes

Reviewer #2: Yes

5. Is the manuscript presented in an intelligible fashion and written in standard English?

Reviewer #1: Yes

Reviewer #2: (No Response)

Reviewer #1: The authors have addressed the comments I made earlier. I have a couple of minor additional suggestions for them to consider.

Intro:

One minor suggestion to update this sentence by inserting something where I put in caps: "By uncovering the specific contextual, demographic, and psychosocial factors [ASSOCIATED WITH ...], this study provides insights into the mechanisms that contribute to the effectiveness of the intervention across diverse populations."

Discussion

In discussion line 298, the authors introduce the term "diagnostic overload". They cite Garg paper, which does talk about how lack of effective intervention can increase inequity. But I don't think it's appropriate to quote "diagnostic overload" here. It doesn't come from Garg paper--I went back to read that paper out of curiosity. Garg's paper refers to the inadequacy of interventions, not diagnostic overload. I think these are subtly different and I would distinguish between them.

Also one very minor grammatical thing I noticed in this sentence below. I think it is missing a dash where I put in brackets: "In

contrast, Class 2 – representing the most disadvantaged group [—] reported higher unmet social care needs

and a persistent trend of increasing needs post-intervention, indicating that more intensive support might

be required beyond the digital intervention alone.

Reviewer #2: Overall satisfied with the responses and the resulting changes to the paper. About my comment on missing data, the response I received needs to be included in the methods section.

**Do you want your identity to be public for this peer review?** For information about this choice, including consent withdrawal, please see our Privacy Policy

Reviewer #1: No

Reviewer #2: No

---

## [Author Response · Author response to Decision Letter 2]

16 Dec 2025

Point-by-point response to Reviewers’ comments

Dear Editor,

We thank you and the reviewers for your time in reviewing our paper and providing valuable comments that have helped to improve the current version. We have carefully considered the comments and responded to all the items. We hope the manuscript after careful revisions meet your high standards. Below we provide the point-by-point responses. All modifications in the manuscript have been highlighted as tracked changes.

Editor Comments

The reviewers appreciated your attention to their suggestions and had only a few minor comments. Please note the request to include a statement about your missing data within the methods section of your manuscript. We would encourage you to resubmit these minor corrections as soon as you can.

Author response: We thank the Editor for the feedback. We have now added a statement about the missing data within the methods section of the manuscript and have also addressed the other feedback raised by the reviewers as below.

Reviewer #1

The authors have addressed the comments I made earlier. I have a couple of minor additional suggestions for them to consider.

1. Intro: One minor suggestion to update this sentence by inserting something where I put in caps: "By uncovering the specific contextual, demographic, and psychosocial factors [ASSOCIATED WITH ...], this study provides insights into the mechanisms that contribute to the effectiveness of the intervention across diverse populations."

Author response: We thank the reviewer for the feedback. We have now revised the sentence as suggested to improve clarity (See Page 6, Lines 109-112).

Modifications: “By uncovering the specific contextual, demographic, and psychosocial factors associated with differential engagement with the WMG-E tool and variation in the identification and support of unmet social care needs, this study provides insights into the mechanisms that contribute to the effectiveness of the intervention across diverse populations.”

2. Discussion

In discussion line 298, the authors introduce the term "diagnostic overload". They cite Garg paper, which does talk about how lack of effective intervention can increase inequity. But I don't think it's appropriate to quote "diagnostic overload" here. It doesn't come from Garg paper--I went back to read that paper out of curiosity. Garg's paper refers to the inadequacy of interventions, not diagnostic overload. I think these are subtly different and I would distinguish between them.

Author response: We agree with the reviewer and we have revised the text to more accurately reflect the argument presented in Garg’s work, namely that screening for social needs in the absence of effective, accessible, and timely intervention pathways may inadvertently exacerbate inequities (See Page 20, Lines 303-312).

Modifications: “The increase in reported social care needs among Class 2 participants likely reflects the identification of unmet needs without sufficient capacity or intensity of intervention to address them, a phenomenon highlighted in prior work examining social risk screening in healthcare settings [26]. In such contexts, screening without effective, accessible, and timely intervention pathways may unintentionally exacerbate inequities, particularly for families facing structural and systemic barriers to care, including limited financial resources, reduced local service availability, transportation challenges, and difficulties balancing work or caregiving responsibilities.”

3. Also one very minor grammatical thing I noticed in this sentence below. I think it is missing a dash where I put in brackets: "In contrast, Class 2 – representing the most disadvantaged group [—] reported higher unmet social care needs and a persistent trend of increasing needs post-intervention, indicating that more intensive support might be required beyond the digital intervention alone.

Author response: We have now added the dash as suggested (See Page 19, Lines 274-277).

Modifications: “In contrast, Class 2 – representing the most disadvantaged group – reported higher unmet social care needs and a persistent trend of increasing needs post-intervention, indicating that more intensive support might be required beyond the digital intervention alone.”

Reviewer #2

1. Overall satisfied with the responses and the resulting changes to the paper. About my comment on missing data, the response I received needs to be included in the methods section.

Author response: We thank the reviewer for the feedback. We have added the sentence about missing data in the data analysis section (See Pages 8-9, Lines 166-168).

Modifications: “All data were complete except for the ‘current service use’ variable, which had three missing cases. Given the minimal extent of missing data, multiple imputation was not undertaken, and analyses were conducted using available-case data.”

---

## [Decision Letter · Decision Letter 2]

23 Dec 2025

Identifying subgroups with differential levels of service response to a digital screening and service navigation program for unmet social care needs

PONE-D-25-48221R2

Dear Dr. Eapen,

We’re pleased to inform you that your manuscript has been judged scientifically suitable for publication and will be formally accepted for publication once it meets all outstanding technical requirements.

Kind regards,

Onaedo Ilozumba

Academic Editor

PLOS One

Reviewers' comments:

Reviewer's Responses to Questions

**Comments to the Author**

Reviewer #2: All comments have been addressed

2. Is the manuscript technically sound, and do the data support the conclusions?

Reviewer #2: Yes

3. Has the statistical analysis been performed appropriately and rigorously?

Reviewer #2: Yes

4. Have the authors made all data underlying the findings in their manuscript fully available?

Reviewer #2: Yes

5. Is the manuscript presented in an intelligible fashion and written in standard English?

Reviewer #2: Yes

Reviewer #2: I am satisfied with the authors' responses and the resulting changes to the paper. I have nothing else to add.

**Do you want your identity to be public for this peer review?** For information about this choice, including consent withdrawal, please see our Privacy Policy

Reviewer #2: No

---

## [Editor Report · Acceptance letter]

PONE-D-25-48221R2

PLOS One

Dear Dr. Eapen,

I'm pleased to inform you that your manuscript has been deemed suitable for publication in PLOS One. Congratulations! Your manuscript is now being handed over to our production team.

Kind regards,

on behalf of

Dr. Onaedo Ilozumba

Academic Editor

PLOS One